# Hygienic behaviors during the COVID-19 pandemic may decrease immunoglobulin G levels: Implications for Kawasaki disease

**Hiromi Yamaguchi[1,2], Masaaki Hirata[1], Kuniya Hatakeyama[1], Ichiro Yamane[1], Hisashi Endo[1], Hiroe Okubo[1], Yoshimi Nishimura[1], Yoshiro Nagao[1] \***

1 Department of Pediatrics and Neonatology, Fukuoka Tokushukai Hospital, Kasuga, Fukuoka, Japan,
2 Department of Pediatrics, Fukuoka University, Fukuoka, Japan

\* ng999214@iis.u-tokyo.ac.jp

**Data Availability Statement:** The data underlying the results presented in the study are available from Kaggle (https://www.kaggle.com/datasets/

## Abstract

### Background

Due to the coronavirus disease 2019 (COVID-19) pandemic, hygienic behaviors became a new norm since January 2020. The hygiene hypothesis predicts that an excessively hygienic environment may adversely affect human health.

### Objective

We quantified the effect of COVID-19 on immunological parameters linked to the hygiene hypothesis.

### Methods

We examined age-specific levels of total nonspecific immunoglobulin G (IgG) and IgE in individuals who visited Fukuoka Tokushukai Hospital between 2010 and 2021. Pre-COVID (2010–2019) and COVID (2020–2021) periods were compared.

### Results

IgG levels steadily decreased throughout Pre-COVID period. IgG levels fell abruptly from the pre-COVID period to the COVID period in all age groups (P = 0.0271, < 0.3 years; P = 0.0096, 0.3–5 years; P = 0.0074, $\geq$ 5 years). The declines in IgG in < 0.3 years and that in $\geq$ 5 years accelerated during the COVID period. IgE levels were seasonal, but did not change noticeably from the pre-COVID to COVID period. IgG levels recorded for patients with Kawasaki disease (KD) (mean 709 mg/dL) were significantly lower than for matched control subjects (826 mg/dL) (P<0.0001).

### Discussion

Hygienic behaviors during the COVID-19 outbreak decreased the chance of infection, which may explain the decreases in IgG levels in children and adults. Neonatal IgG declined, possibly because of the decrease in maternal IgG.

yoshironagao/hygienic-behaviors-during-the-covid-stata-files).

**Funding:** The author(s) received no specific funding for this work.

**Competing interests:** The authors have declared that no competing interests exist.

## Conclusion

Hygienic behaviors decreased the IgG levels in all age groups, from neonates to adults. This downturn in IgG may lead to vulnerability to infections as well as to KD.

## Introduction

The new variant coronavirus (SARS-CoV-2) was reportedly brought to Japan in January, 2020 [1]. Following the global and domestic spread of coronavirus disease 2019 (COVID-19), the Japanese public voluntarily took precautional measures, such as wearing face coverings, and distancing from each other. In March 2020, 67% of the Japanese wore face masks in public places, at a higher rate than in Western countries (e.g. 42% in Spain, 17% in the US and 6% in the UK) [2]. The Japanese government declared a state of emergency repeatedly and requested that the public refrain from non-urgent travel and gatherings. More rigorous regulations were enforced in populated prefectures including Fukuoka prefecture (population 5 million), where our hospital is located.

It is hypothesized that the strict hygiene regulations practiced during the COVID-19 pandemic may affect microbiota that inhabit humans and cause immunological problems [3]. This concern is based upon the long-held hypothesis that as the environment becomes more hygienic, decreasing exposure to infections can adversely affect human health [4]. One study found that hygienic conditions are correlated with an increased risk for inflammatory bowel disease [5]. It was suggested that targeted hygiene against pathogens and sharing essential microbes are both important [6].

Serum Immunoglobulin E (IgE) is an indicator of allergic propensity. Numerous studies revealed that insufficient exposure to microbial diversity during the early days of life induces high IgE levels, which subsequently leads to autoimmune and allergic disorders [7–14]. In contrast to IgE, whose role in protective immunity is not fully understood, Immunoglobulin G (IgG) is a critical mediator of infection immunity. However, there are few studies of IgG in the context of the hygiene hypothesis. In wild animals, nonspecific total IgG levels are higher than in captive animals [15, 16]. Nonspecific total IgG, which is frequently measured in laboratory tests in Japan, could potentially be a useful indicator of nonspecific infection immunity.

The hygiene hypothesis has been proposed as a possible explanation for Kawasaki disease (KD) [17–19]. KD is a febrile pediatric illness with mucocutaneous manifestations. KD most frequently affects children < 5 years of age in Japan and in many other regions/countries [20–22]. Importantly, KD is associated with high rates of potentially fatal coronary complications [23–25]. Although KD was first reported in 1967 [26], its etiology remains unknown. Numerous study results suggest that KD has an infectious etiology. For example, in Japan, KD incidence has a bimodal seasonality that is identical to pediatric viral infections [27]. KD is rare at < 3 months of age [28], which indicates the presence of maternal immunity. KD cases cluster in space and time [29–31]. B cell selection in patients with KD is consistent with an infectious origin rather than an autoimmune origin [32]. Meanwhile, a genetic preposition to KD has been found [33–36]. Therefore, it is widely accepted that KD is triggered when a genetically susceptible individual is infected by an as yet undetermined microbe(s) [37].

In this study, we examined whether the rigorous hygiene regulations imposed during the COVID-19 pandemic affected immunity in the human population. We also attempted to develop a hypothesis about the relationship between the change in population immunity and health outcomes, in particular for KD epidemiology.

## Results

### Human movement and distancing in the study area

Fig 1 shows that people in Fukuoka prefecture avoided public places and workplaces to the greatest degree through the first half of 2020. This avoidance culminated in May 2020 with a > 50% reduction in visits to transit stations. Throughout 2020 and 2021, the degree of this indicator fluctuated, but remained 20% below baseline. People started pre-emptive distancing well before states of emergency were declared.

### Measurement of IgG and IgE

Between January 2010 and December 2021, total nonspecific IgG and IgE were tested 33,107 and 6,530 times, respectively. To avoid the effects of duplicate sampling on the analysis, we used only the first measurement for each individual (19,744 and 5,433 individuals for IgG and IgE, respectively. Table 1).

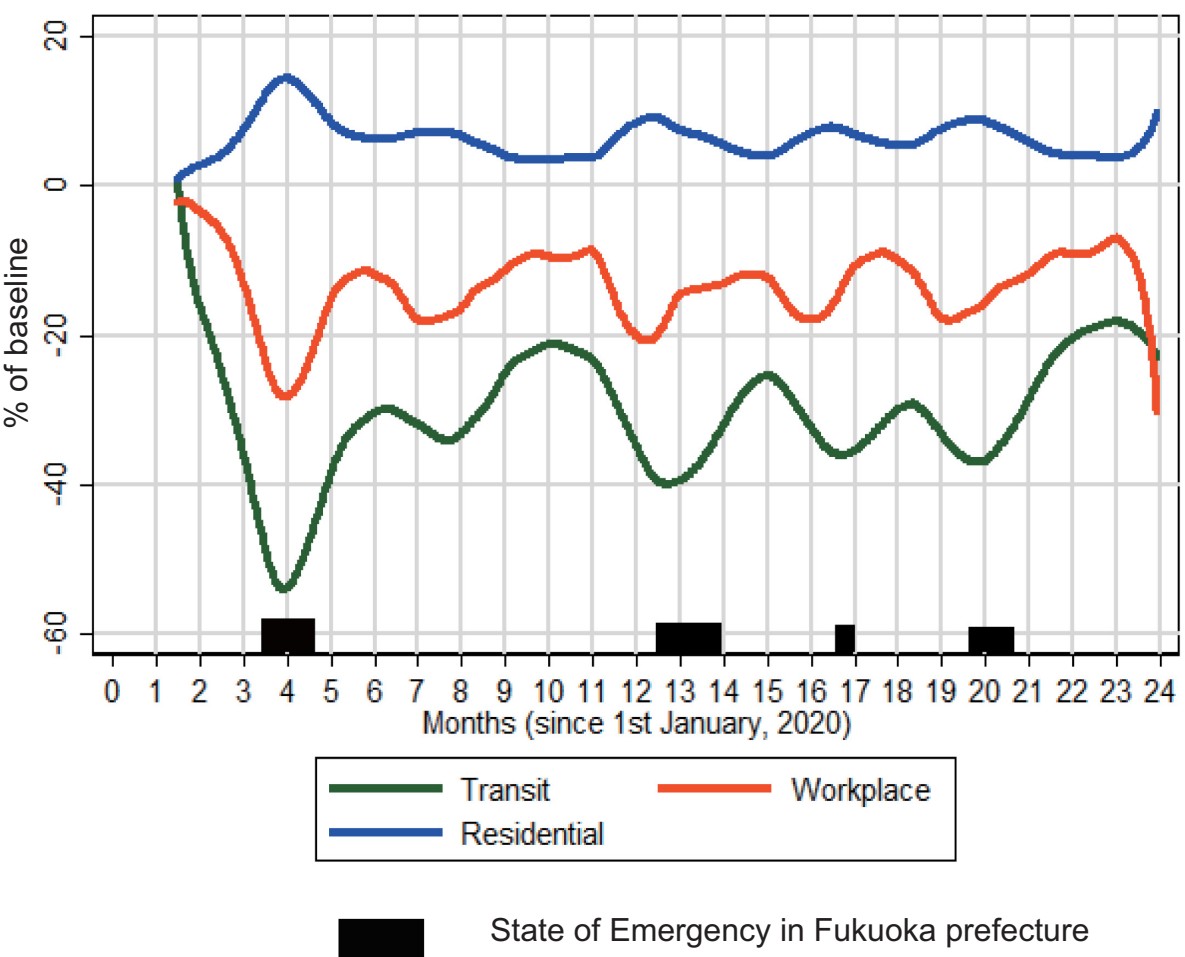

**Fig 1. Indicators of community mobility in Fukuoka prefecture.** Daily data for Fukuoka prefecture was downloaded from the Community Mobility Report [38]. Time spent in residential areas, workplaces, and transit stations were represented as percentages compared to baseline (3 January– 6 February 2020); values were smoothed using the Lowess method and a 3-day bandwidth.

**Table 1. Numbers of individuals for whom total nonspecific IgG or IgE were measured between 2010 and 2021.**

| Year | IgG | | | | IgE | | | |
|------|-------|--------|------|-------|-------|--------|--------|-------|
|      | < 0.3 y | 0.3–5 y | ≥ 5 y | Total | < 5 y | 5–20 y | ≥ 20 y | Total |
| 2010 | 660 | 133 | 473 | 1266 | 90 | 111 | 105 | 306 |
| 2011 | 583 | 73 | 454 | 1110 | 126 | 82 | 236 | 444 |
| 2012 | 650 | 122 | 681 | 1453 | 269 | 109 | 244 | 622 |
| 2013 | 643 | 167 | 535 | 1345 | 318 | 74 | 128 | 520 |
| 2014 | 688 | 189 | 725 | 1602 | 268 | 103 | 129 | 500 |
| 2015 | 743 | 314 | 873 | 1930 | 243 | 105 | 133 | 481 |
| 2016 | 807 | 361 | 1063 | 2231 | 190 | 88 | 92 | 370 |
| 2017 | 727 | 249 | 865 | 1841 | 169 | 82 | 87 | 338 |
| 2018 | 602 | 326 | 699 | 1627 | 177 | 69 | 171 | 417 |
| 2019 | 613 | 322 | 784 | 1719 | 140 | 89 | 228 | 457 |
| 2020 | 413 | 229 | 980 | 1622 | 101 | 78 | 246 | 425 |
| 2021 | 353 | 348 | 1297 | 1998 | 144 | 91 | 318 | 553 |
| Total | 7482 | 2833 | 9428 | 19744 | 2235 | 1081 | 2117 | 5433 |

## IgG and age

IgG level had a mostly positive correlation with age (Spearman's R = 0.3962, P < 0.0001, n = 19,744, Fig 2A). IgG levels declined from birth to 3–4 months of age; this change reflected waning maternal immunity (Fig 2B). Subsequently, IgG levels steeply surged up to 5 years of age. Based on this result, in the analysis of IgG we classified individuals into three age groups: infants and neonates < 0.3 years; children between 0.3 and 5 years; individuals ≥ 5 years.

## Temporal shift in IgG

Annual IgG levels declined over the years in all age groups (Fig 3). There was a statistically significant decrease in IgG from 2019 to 2020 in all age groups (P = 0.0096 in < 0.3 y; P = 0.0271 between 0.3 and 5 y; P = 0.0074 in ≥ 5 y, Mann-Whitney-Wilcoxon test) (Fig 3A). Fig 3(B) presents this result using a finer temporal resolution; IgG declined dramatically throughout 2020 in early infants (< 0.3 year) and in individuals > 5 years. These findings were supported by the results of regression analyses (Table 2): the downward temporal trends in IgG were statistically significant in all age three groups during the pre-COVID period (i.e., 2010–2019). In contrast, the decline in IgG was significant only in early infants and neonates (< 0.3 year) and individuals ≥ 5 years during the COVID period (i.e., 2020–2021). This temporal trend was significant even after including age in the multivariate analysis. The results for the multivariate coefficients (Table 2) indicated that IgG decreased by 2.0 mg/dL (< 0.3 years), 9.2 mg/dL (0.3–5 years), and 14 mg/dL (≥ 5 years) per year, during the pre-COVID period. During the COVID period, IgG declined annually by 43 mg/dL (< 0.3 years) and 48 mg/dL (≥ 5 years). IgG did not decrease significantly in the 0.3–5 years age group during the COVID period. We repeated these analyses after transforming IgG data into a normal distribution using the Box-Cox method [39], and qualitatively validated the results (S1 Table). Subdivision of the age class ≥5 years showed that, during the COVID period, the IgG declined by 42.1 mg/dL per year in 5–50 years age group (P = 0.0138, n = 1,325) and 42.5 mg/dL per year in ≥50 years (P = 0.1521, n = 952) (S2 Table).

The results presented in Fig 3(B) suggested that IgG fluctuated over time. Periodicity analysis found that the cycle of this periodicity was much longer than 2.5 years (S1 Fig). Because the study period was only 12 years, we did not consider effects of this fluctuation in the statistical analyses.

(a) Entire ages

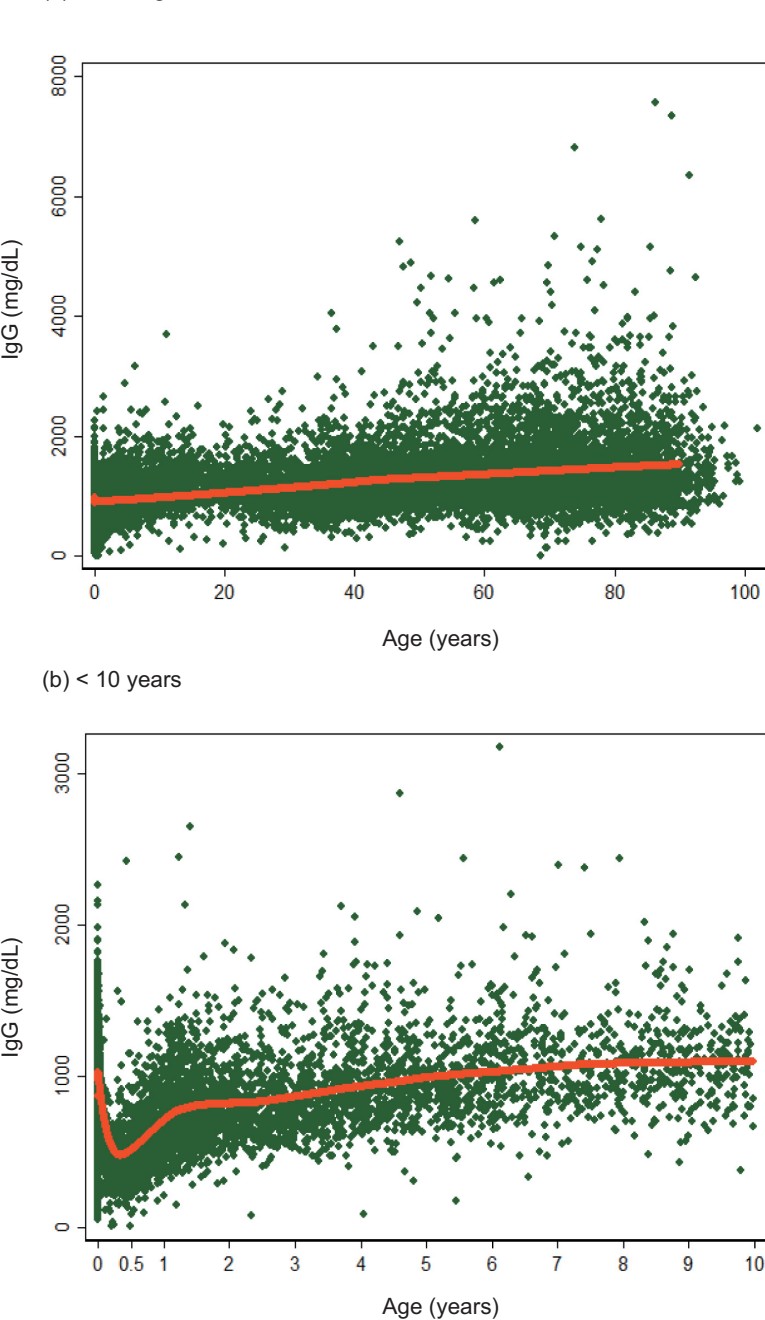

(b) < 10 years

**Fig 2. IgG and age.** Nonspecific total IgG values, which were tested between January 2010 and December 2021 in Fukuoka Tokushukai Hospital, were plotted against ages of individuals, for all age groups (a), and for individuals < 10 years of age (b). The red lines represent the results for Lowess smoothing with a 14-day bandwidth.

## IgE and age

The correlation of IgE values with age was significant (Spearman's R = 0.5368, P<0.0001, n = 2,988). IgE is not transferred transplancentally. IgE steeply increased as age increased, up to 20 years of age. Therefore, for the analyses of IgE, individuals were classified into three age classes: < 5 years, 5–20 years, and ≥ 20 years.

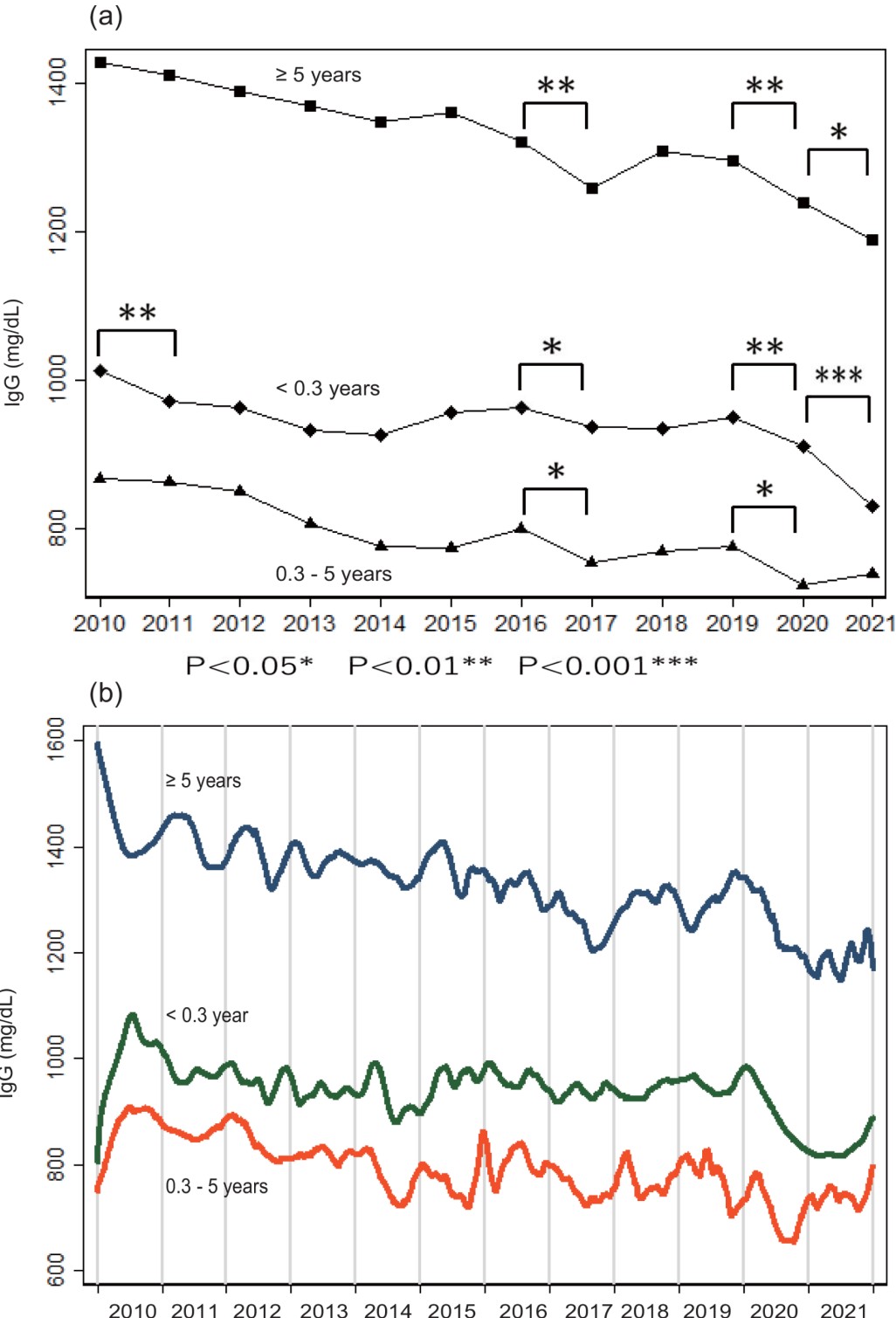

**Fig 3. Temporal changes in IgG.** (a) IgG values averaged for each year are presented for three age groups of individuals of < 0.3 years (diamond), 0.3–5 years (triangle) and ≥ 5 years (square). Statistical comparisons between IgG values reported in one year and those reported in an adjacent year were performed using sum of rank tests (i.e., Mann-Whitney-Wilcoxon test). * P<0.05, ** P<0.01, *** P<0.001. (b) IgG values from these three age groups were smoothed using the Lowess method with a 14-day bandwidth.

**Table 2. Linear regression coefficients to explain IgG in different age groups.**

| Period | < 0.3 years | | 0.3–5 years | | ≥ 5 years | |
|---|---|---|---|---|---|---|
| 1. Pre-COVID (2010–2019) | n = 6716 | | n = 2256 | | n = 7152 | |
| 1.1 Univariate | Coefficient | P | Coefficient | P | Coefficient | P |
| Time* (years) | - 4.81 | P = 0.0002 | - 10.0 | P<0.0001 | - 16.8 | P<0.0001 |
| Adjusted R² | 0.1843 | P = 0.0002 | 0.0077 | P<0.0001 | 0.0070 | P<0.0001 |
| 1.2. Multivariate | Coefficient | P | Coefficient | P | Coefficient | P |
| Time (years) | - 2.01 | P = 0.0895 | - 9.2 | P<0.0001 | - 13.6 | P<0.0001 |
| Age† (years) | - 2770 | P<0.0001 | 94 | P<0.0001 | 5.18 | P<0.0001 |
| Adjusted R² | 0.1784 | P<0.0001 | 0.1841 | P<0.0001 | 0.0656 | P<0.0001 |
| 2. COVID (2020–2021) | n = 766 | | n = 577 | | n = 2277 | |
| 2.1 Univariate | Coefficient | P | Coefficient | P | Coefficient | P |
| Time (years) | -82.1 | P<0.0001 | 4.23 | P = 0.8261 | - 48.8 | P = 0.0039 |
| Adjusted R² | 0.0305 | P<0.0001 | -0.0017 | P = 0.8261 | 0.0032 | P = 0.0039 |
| 2.2. Multivariate | Coefficient | P | Coefficient | P | Coefficient | P |
| Time (years) | - 42.6 | P = 0.0029 | 13.1 | P = 0.4395 | -48.1 | P = 0.0029 |
| Age (years) | - 2455 | P<0.0001 | 99.4 | P<0.0001 | 5.14 | P<0.0001 |
| Adjusted R² | 0.3096 | P<0.0001 | 0.2292 | P<0.0001 | 0.0849 | P<0.0001 |

*Time that elapsed from the beginning of study period to the date of IgG measurement (in years).

† Age at the date of IgG measurement. For both time and age, the minimal temporal resolution used was day, while the unit of time is expressed in year.

## Temporal shift in IgE

When IgE levels were smoothed over the time in the analysis, there was seasonality in all age classes (Fig 4). To identify periodicities, we performed periodogram analysis of each age class (Fig 5). Periodicity with a one-year cyclicity (i.e., seasonality) was found in children < 5 years of age. However, seasonality was more obscure in the older classes. We used regression analysis to identify temporal trends in IgE. A linear combination of sine and cosine transformations of time was included in multivariate analyses to represent seasonality. Only the results of the multivariate analysis performed for the < 5 years age class and in the pre-COVID period indicated that there was a significantly negative temporal trend (Table 3). A multivariate analysis revealed that the sine term had a significant correlation with IgE, which indicated the presence of seasonality. We repeated these regression analyses by transforming IgE values into a normal distribution [39], and validated the results qualitatively (S3 Table). Taken together, we did not find unequivocally that COVID-19 affected IgE in the population (S2 Fig).

## Comparison of IgG between patients with KD and matched control individuals

IgG levels were compared between 314 children with KD and an equivalent number of matched control children (Fig 6). In both groups, the prevalence of female subjects was 40.8%. The mean age at IgG testing was 2.63 years for the group of patients with KD and 2.67 years for the matched control group. The mean IgG was 709 mg/dL (95% confidence interval: 682–736 mg/dL) for the group of patients with KD and 826 mg/dL (789–862 mg/dL) for the matched control group (P<0.0001, Wilcoxon's matched-pairs signed-ranks test).

To consider the possibility that this lower IgG in KD patients may have biased our previous analysis, we repeated the regression analysis (Table 2) by excluding the patients with KD. However, the result was not affected qualitatively (S4 Table).

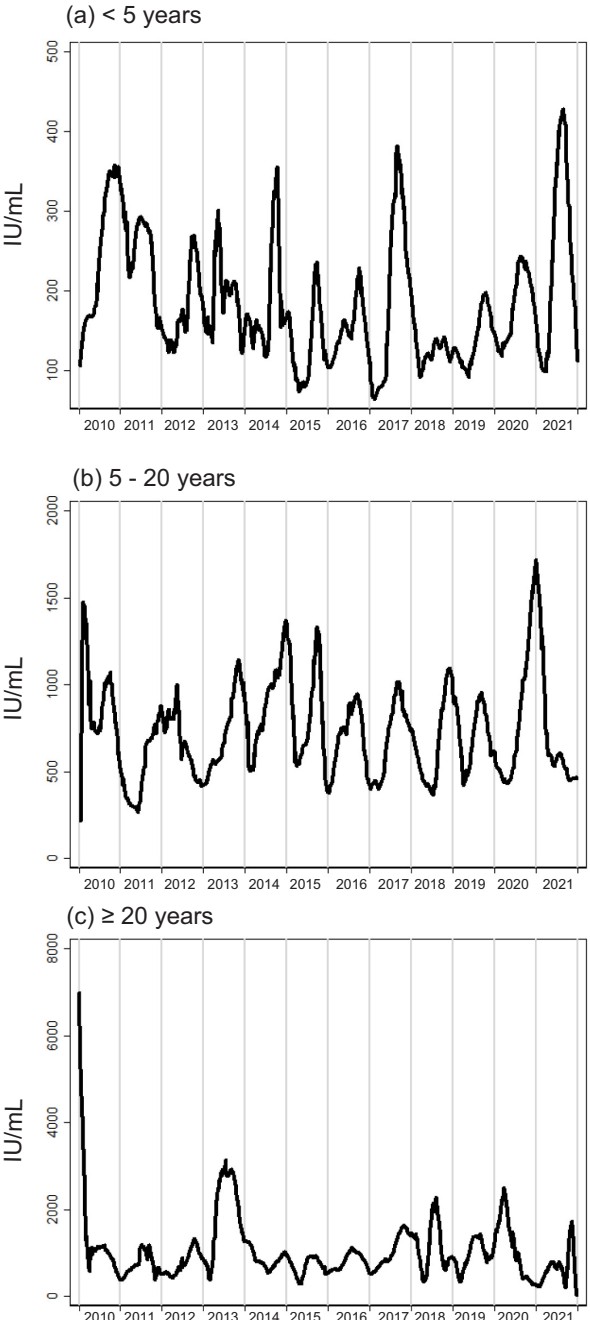

**Fig 4. Temporal changes in IgE.** IgE values were recorded for each age group, (a) < 5 years, (b) 5–20 years, and (c) ≥ 20 years, and were smoothed using the Lowess method (bandwidth = 14 days).

## Discussion

We examined whether changes in hygiene behavior during the COVID-19 outbreak affected nonspecific immunity represented by total IgE and IgG. IgE levels presented seasonality, being consistent with previous reports [40, 41]. However, there was no unambiguous result which supported presence of temporal trend in IgE or effect of COVID-19 on IgE levels. In contrast,

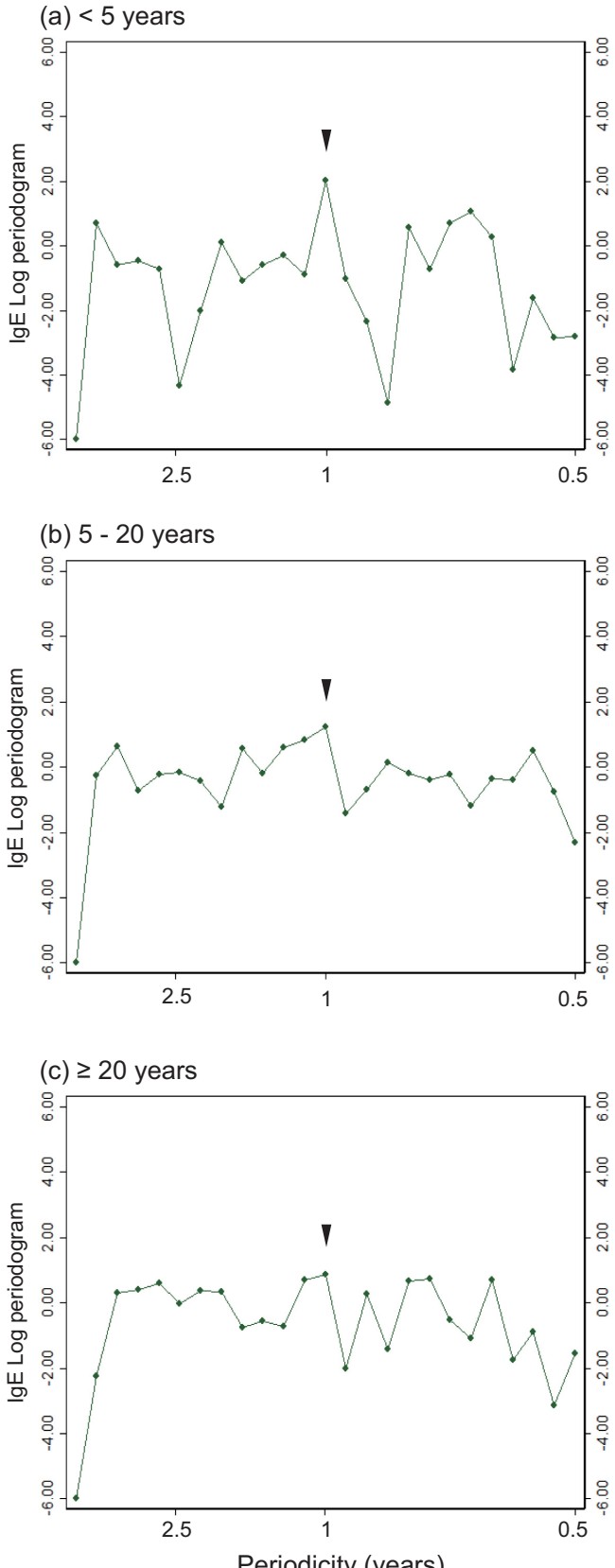

**Fig 5. Periodicity of IgE.** IgE values were averaged for each quarter of the year, in each age group. A periodogram was generated from this time-series data. A periodicity with a cycle of one year (i.e., seasonality) was found (arrow heads) for children < 5 years of age (a). This seasonality was more obscure in individuals between 5 and 20 years of age (b), and < 20 years of age (c).

IgG decreased slowly but steadily since at least 2010. The rate of decrease in IgG abruptly accelerated in 2020, coinciding with the COVID-19 pandemic. This drop in IgG was most prominent in populations > 5 years of age and in neonates and early infants < 0.3 years of age. These results suggested that hygienic behaviors (e.g., face coverings and distancing) reduced opportunities for infection with microbes, leading to the decline in IgG. Because the mean half-life of IgG is 21 days [42], lack of infection would most likely decrease the IgG without delay. The mother's IgG is transferred to the fetus during the last few months of pregnancy [43]. The IgG levels in mothers and in umbilical cord blood are highly correlated [44]. Therefore, a downturn in maternal IgG would be reflected in a change in neonatal IgG. In contrast, children between 0.3 years and 5 years would likely not have been affected by the reduction in infection because they were not required to wear face coverings during the COVID-19 outbreak in Japan. Interestingly, there was a significant IgG decline in all age groups from 2016 to 2017 (Fig 3A). This decrease may have been related to an increase of the face mask production in Japan which occurred in 2015 [45].

The KD incidence in Japan decreased by 35% in 2020 from the pre-COVID years [46], implying that the hygiene behaviors may have affected this illness. Therefore, it would be worthwhile to discuss the potential impact of the decreasing IgG on KD. The number of KD cases has been increasing rapidly in Japan and in many developed countries/regions [47, 48].

**Table 3. Regression analysis to explain IgE in different age groups.**

| | < 5 years | | 5–20 years | | ≥ 20 years | |
|---|---|---|---|---|---|---|
| 1. Pre-COVID (2010–2019) | n = 1990 | | n = 912 | | n = 1553 | |
| 1.1 Univariate | Coefficient | P | Coefficient | P | Coefficient | P |
| Time (year) | -7.47 | P = 0.1285 | -9.97 | P = 0.5106 | 3.93 | P = 0.9009 |
| Adjusted $R^2$ | 0.0007 | P = 0.1285 | -0.0006 | P = 0.5106 | -0.0006 | P = 0.9009 |
| 1.2. Multivariate | Coefficient | P | Coefficient | P | Coefficient | P |
| Time (year)* | -10.1 | P = 0.0335 | -12.0 | P = 0.4287 | 5.46 | P = 0.8628 |
| Age (year) † | 113 | P<0.0001 | 33.0 | P = 0.0052 | -5.14 | P = 0.3134 |
| $\sin(2\pi \times Time)$ | -57.8 | P = 0.0005 | -158 | P = 0.0087 | -297 | P = 0.0257 |
| $\cos(2\pi \times Time)$ | -19.0 | P = 0.2693 | 34.8 | P = 0.5828 | -250 | P = 0.0726 |
| Adjusted $R^2$ | 0.0756 | P<0.0001 | 0.0125 | P = 0.0039 | 0.0030 | P = 0.0721 |
| 2. COVID (2020–2021) | n = 245 | | n = 169 | | n = 564 | |
| 2.1 Univariate | Coefficient | P | Coefficient | P | Coefficient | P |
| Time | 106 | P = 0.1704 | -69.3 | P = 0.7262 | -304 | P = 0.2625 |
| Adjusted $R^2$ | 0.0036 | P = 0.1704 | -0.0052 | P = 0.7262 | 0.0005 | P = 0.2625 |
| 2.2. Multivariate | Coefficient | P | Coefficient | P | Coefficient | P |
| Time (year)* | 33.3 | P = 0.6663 | -187 | P = 0.3660 | -235 | P = 0.4063 |
| Age (year) † | 143 | P<0.0001 | 108 | P = 0.0002 | -9.93 | P = 0.2000 |
| $\sin(2\pi \times Time)$ | -133 | P = 0.0246 | -146 | P = 0.3710 | 174 | P = 0.4367 |
| $\cos(2\pi \times Time)$ | -107 | P = 0.0697 | 320 | P = 0.0410 | -48.9 | P = 0.8167 |
| Adjusted $R^2$ | 0.0965 | P<0.0001 | 0.0779 | P = 0.0017 | -0.0009 | P = 0.4771 |

*Time elapsed from the beginning of study period to the date of IgE measurement (by year).

† Age at date of IgE measurement. For both time and age, the minimal temporal resolution was day, while the unit of time is expressed in year.

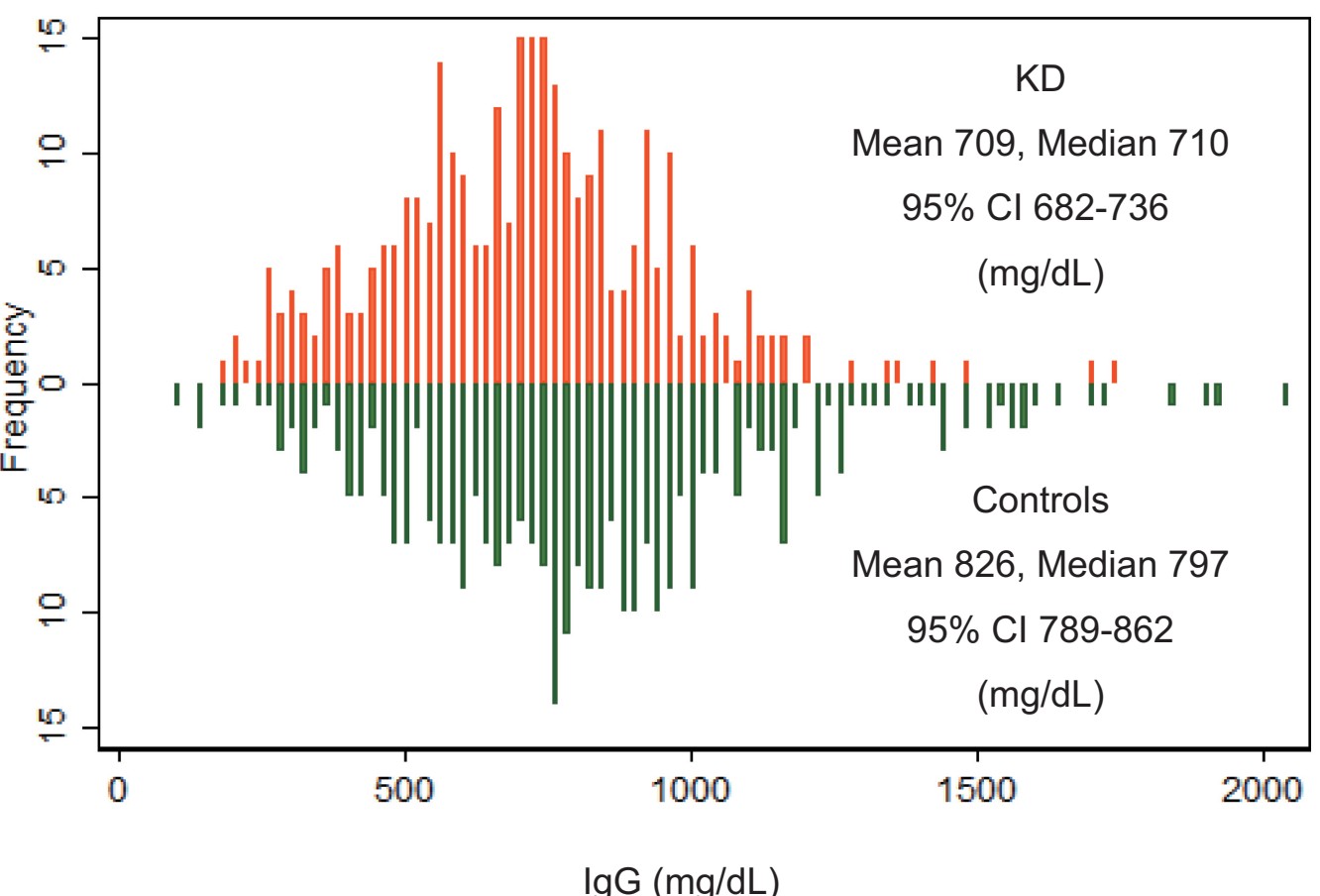

**Fig 6. IgG distributions of patients with KD and matched control individuals.** Patients with KD were selected from the pre-COVID period (2010–2019). Each control individual was matched to a patient with KD in terms of sex, age, and date of IgG testing. IgG values in patients with KD were significantly lower than those in control individuals.

This constant increase in developed countries prompted discussion about the hygiene hypothesis [17]. Consistent with the hypothesis, KD risk is positively correlated with higher income, urbanization, and smaller family size [18, 49]. Lee hypothesized that the etiology of KD is dysregulated early B cell development under reduced microbial exposure [19]. Study findings indicate that before intravenous immunoglobulin (IVIG) administration, serum IgG levels in patients with KD are lower than in individuals without KD [50]. To our knowledge, our study was the first to quantify this phenomenon using accurately matched control subjects. Whether this low IgG level in patients with KD is a consequence of the systemic inflammation associated with KD or a predisposing factor for development of the illness, or both, remains unknown. Although IVIG is the mainstay of KD treatment, the mechanism for how IVIG cures KD is unknown. It was hypothesized that the IgG in IVIG suppresses excessive immune reactions or neutralizes causative microbes, or both [51]. A low serum IgG level before the first IVIG treatment predicts unresponsiveness to the treatment [50, 52], possibly because IVIG fails to sufficiently elevate the IgG level [53]. A lower level of IgG after the initial IVIG administration predicts a risk of coronary aneurysm [54–56]. Therefore, it is likely that a child with

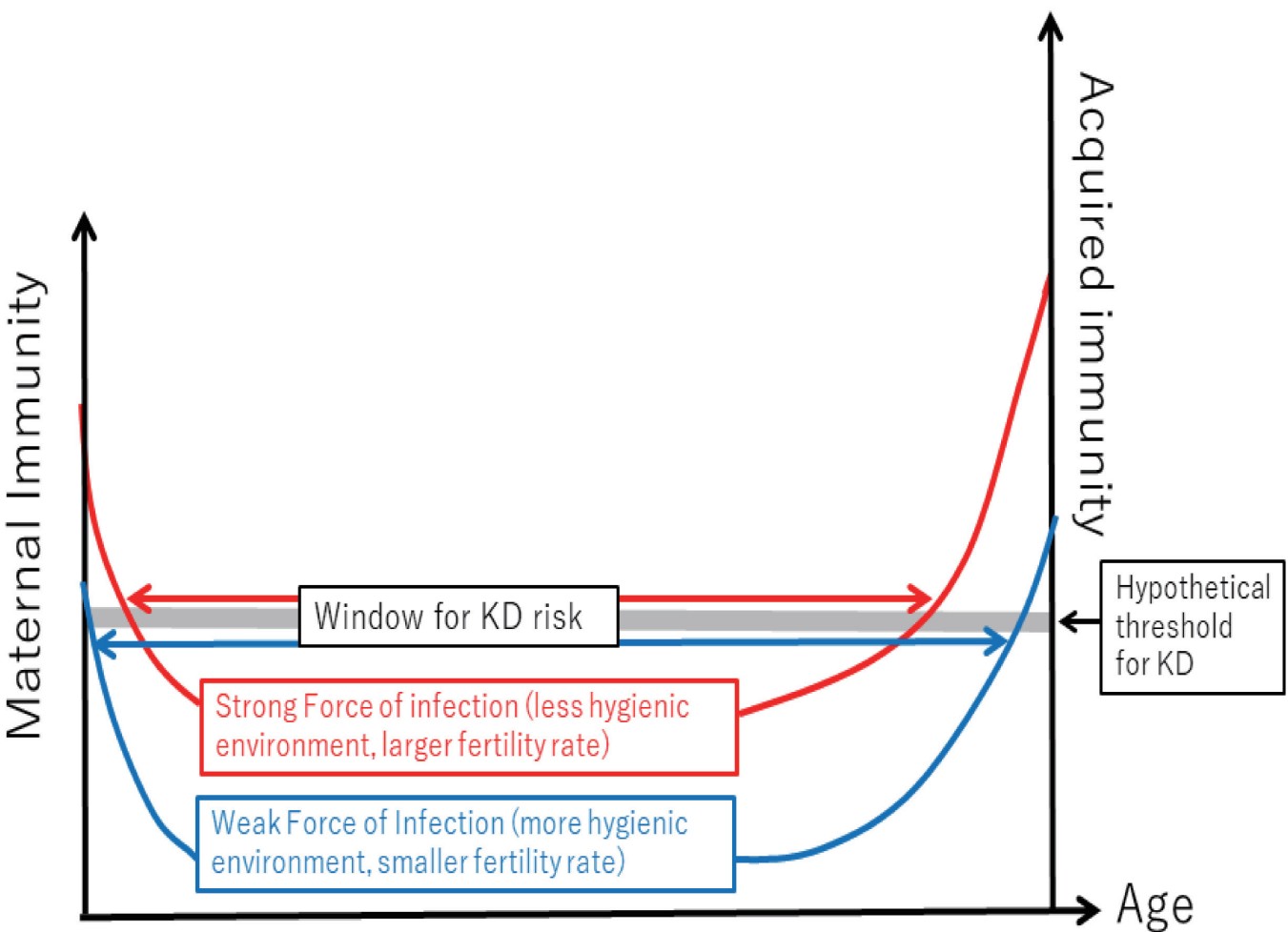

**Fig 7. Hypothesis that explains and predicts expansion of ages of KD due to decreased IgG levels.** The force of infection is high in a less hygienic environment (red). As the environment becomes more hygienic, the force of infection becomes lower (blue). Under an assumption that KD occurs more frequently under a hypothetical threshold of IgG, the age window for KD risk expands into both younger and older ages.

lower IgG may be more prone to develop KD (or at least complicated KD) than children with higher IgG, when triggered by one or more unknown aetiologic agents. This hypothesis may explain why preterm birth is associated an elevated risk for KD [57], because IgG levels in preterm babies are lower than in term babies (S3 Fig) [43]. In a sense, the "window" of pediatric ages at which children are vulnerable to KD (or at least IVIG-refractory KD) may correspond to a hypothetical threshold of IgG (Fig 7). This study revealed that IgG levels in all age groups constantly decreased over at least a decade, and this downward trend accelerated during the COVID-19 outbreak. This finding suggested that the "window" of age vulnerable to KD expands into both older and younger age groups. This mechanism may also explain why the age distribution of KD in Japan has expanded in both directions since 1970 [49]. In 2020, the number of patients with KD decreased in Japan [46, 58]. However, our window hypothesis predicts that the incidence of KD may increase after a transient decrease, in a way that has been observed for other infectious diseases [59].

The results of one study suggested that a decline in the total fertility rate in Japan caused an increase in KD incidence, with a lag of 15 years [49] (S4 Fig). Total fertility rate has also been

found to be a surrogate for force of infection of pediatric infectious diseases [27]. Based on these findings, we hypothesized that a decline in the force of infection increased KD incidence after a long delay. The rate of decrease in IgG predicted in this study may seem too small to cause any change in KD epidemiology. However, this very slow rate may explain why it takes 15 years for a decline in fertility rate (and hence, in the force of infection) to increase KD incidence [49].

The new hygienic norm has become prevalent in developed and developing countries. This behavioral change may decrease IgG levels in diverse regions, and result in unexpected consequences for human health. For example, KD may be expanded into previously unaffected regions and ages. Because of decreasing IgG in children, the current standard dose of IVIG (2 g/kg) may become insufficient for an increasing number of patients with KD.

This study had some limitations. The data were derived retrospectively from a single hospital and may not necessarily represent a general, healthy population. The time elapsed during the COVID-19 period was only 2 years. This short period limited the statistical power to detect any temporal trends from this period or to identify a change in the clinical picture of KD, compared with the pre-COVID era. In our dataset, 50.1% of infants < 0.3 years of age were neonates admitted to the neonatal intensive care unit (NICU). However, the proportion of extremely preterm and/or low birthweight babies admitted to the NICU steadily decreased in recent years (S5 Fig). Because these babies have had a limited duration to receive maternal IgG transplacentally [43], their IgG levels are low (S3 Fig). Therefore, the analysis most likely underestimated the rate in IgG decrease in the population < 0.3 years of age. Despite these caveats, we identified a significant downturn in the IgG in this youngest population. This result supported the robustness of our result.

Although indiscriminate extinction of commensal organisms and common harmless microbes would likely induce unfavorable effects on regulatory immunity [60], targeted hygiene should still be maintained [61]. The magnitude of distancing varies greatly between countries/regions. These heterogeneities provide an opportunity to correlate socio-behavioral changes to human immunological parameters (e.g. total IgG) and health outcomes. The results may provide information that can be used to develop effective targeted hygiene measures. Future multi-region long-term studies are warranted to predict the effects of the global COVID pandemic on health problems, including KD.

## Materials and methods

### Ethics statement

This study was approved by the Ethics Committee for Fukuoka Tokushukai Hospital, with reference 220302. The Ethics Committee waived the requirement for informed consent because of the retrospective nature of this study. All data were fully anonymized before the analysis.

### Community mobility

Indicators for time spent in categorized locations, which started 15 February 2020, were downloaded from "COVID-19 Community Mobility Reports–Google" [38]. The baseline represented a median value for day of the week, recorded between 3 January and 6 February 2020. We examined three categories for location (i.e., transit stations, workplace, and residential area), smoothed over time using the Lowess method.

### Study site and period

We extracted clinical records and laboratory data records from the electronic medical recording system of Fukuoka Tokushukai Hospital. This system began operation in August 2009. We

used the dates from January 2010 to December 2021. Medical records for 896,381 individuals were stored in the system on December 2021.

## IgG measurement

IgG was measured using a Clinical Analyzer 7180Ⓡ (Hitachi High-Tech, Japan) until August 2014. A TBA-c16000Ⓡ analyzer (Canon Medical Systems, Japan) was used thereafter. Measurement accuracy was validated once per year by the Medical Society of Fukuoka Prefecture (MSFP). At each validation, the laboratory of our hospital measured two to three samples delivered from the MSFP and returned the values to the MSFP. During the study period, our hospital passed all the validations with "A-level" scores. This result indicated that the results were within 2 standard deviations of values reported from all hospital laboratories in Fukuoka prefecture.

## IgE measurement

Total nonspecific IgE was measured by BML Inc. (Tokyo, Japan) until March 2015, and by the SRL Corporation (Tokyo, Japan) thereafter.

## Regression analysis to detect temporal trends

We used univariate or multivariate linear regression analysis to identify temporal trends in IgG and IgE. Day was the level of resolution for the age and time variables, while unit of time is expressed in year.

## Statistical test to compare IgG between adjacent years

We compared IgG and IgE values between adjacent years (e.g., 2009 vs 2010) using Mann-Whitney-Wilcoxon tests.

## Periodicity analysis

The values for IgG or IgE were averaged for each quarter of each year, in each age class. The resulting time-series data were analyzed using the periodogram method to identify periodicity [62].

## Comparison of IgG between patients with KD and matched control children

We screened 476 patients with KD who had their first clinical diagnosis for KD, between 2010 and 2019 (i.e., pre-COVID era), and for whom IgG was tested before initiation of IVIG treatment. Here, we selected patients with KD from only the pre-COVID period to exclude possible SARS-CoV-2-induced illness, which mimics KD [63–65]. IgG levels were compared between children with KD and the control children. A control child without KD was matched to a case with KD, in terms of sex, age, and date of IgG measurement. The difference in age and that in date of IgG measurement between a patient with KD and a matched control subject were both within 6 months. Using in-house software [66], we matched 314 patients with KD to an equivalent number of control children.

## Statistical software

We used Stata/SE 13.1 (TX, USA) for the statistical analysis. Statistical significance was defined as P<0.05.

### Datasets

The datasets used in the present study are available from https://www.kaggle.com/datasets/yoshironagao/hygienic-behaviors-during-the-covid-stata-files.

## Supporting information

**S1 Fig. IgG periodicity.** IgG values were averaged for each quarter of each year between 2010 and 2021, for each age group. Periodograms were created for these time-series datasets. Seasonality (i.e., one-year cycles) was not found for each age group.
(EPS)

**S2 Fig. Temporal change in IgE.** IgE values were averaged for each year, in individual age groups. IgE values recorded in one year were compared to those in the adjacent year using rank sum tests (i.e., Mann-Whitney-Wilcoxon tests).
(EPS)

**S3 Fig. IgG and gestational age in neonatal intensive care unit (NICU).** Between 2010 and 2021, 3,739 neonates were admitted to the NICU. Among these neonates, this number includes 3,638 neonates for whom IgG was measured within 7 days from birth. IgG was highly correlated with gestational age (adjusted $R^2 = 0.5447$, P<0.0001).
(EPS)

**S4 Fig. Temporal negative correlation between total fertility rate and KD incidence.** The annual incidence of KD reported from 47 prefectures in Japan was correlated strongly with the total fertility rate (TFR) recorded 15 years previously (a). In a prefecture where KD incidence was high, the TFR was low 15 years previously (b). Ref. [49] was modified.
(EPS)

**S5 Fig. Birthweights and gestational ages in neonatal intensive care unit of Fukuoka Tokushukai Hospital.** The mean values for birthweights and gestational ages in neonates who were admitted to the neonatal intensive care unit (NICU) increased. This increase was due to a change in NICU policy.
(EPS)

**S1 Table. Linear regression coefficients to explain normalized IgG in different age groups.**
(DOCX)

**S2 Table. Liner regression coefficients to explain IgG in four age groups.**
(DOCX)

**S3 Table. Regression analysis to explain normalized IgE in different age groups.**
(DOCX)

**S4 Table. Regression coefficients which explain IgG, after excluding Kawasaki disease cases.**
(DOCX)

## Acknowledgments

**Funding sources**

We are grateful to Teppei Nagai, Kazuma Shibata, Akihito Hori, Shinichiro Yamauchi and Hiroshi Manabe for their assistance in data preparation and statistical analysis. There was no funding source.

## Author Contributions

**Conceptualization:** Hiromi Yamaguchi, Yoshiro Nagao.

**Data curation:** Hiromi Yamaguchi.

**Formal analysis:** Yoshiro Nagao.

**Investigation:** Masaaki Hirata, Ichiro Yamane, Hisashi Endo, Hiroe Okubo.

**Methodology:** Kuniya Hatakeyama, Yoshimi Nishimura.

**Resources:** Masaaki Hirata.

**Supervision:** Masaaki Hirata.

**Validation:** Kuniya Hatakeyama.

**Writing – original draft:** Hiromi Yamaguchi, Yoshiro Nagao.

**Writing – review & editing:** Masaaki Hirata, Kuniya Hatakeyama, Ichiro Yamane, Hisashi Endo, Hiroe Okubo, Yoshimi Nishimura.

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
