## [Decision Letter · Decision Letter 0]

27 Jul 2022

PONE-D-22-10882Hygienic behaviors during the COVID-19 pandemic may decrease immunoglobulin G levels: implications for Kawasaki diseasePLOS ONE

Dear Dr. Nagao,

Thank you for submitting your manuscript to PLOS ONE. After careful consideration, we feel that it has merit but does not fully meet PLOS ONE’s publication criteria as it currently stands. Therefore, we invite you to submit a revised version of the manuscript that addresses the points raised during the review process.

We look forward to receiving your revised manuscript.

Kind regards,

Ghulam Md Ashraf, Ph.D.

Academic Editor

PLOS ONE

Journal Requirements:

5. We note that Figures 8 and S5 in your submission contain [map/satellite] images which may be copyrighted. All PLOS content is published under the Creative Commons Attribution License (CC BY 4.0), which means that the manuscript, images, and Supporting Information files will be freely available online, and any third party is permitted to access, download, copy, distribute, and use these materials in any way, even commercially, with proper attribution. For these reasons, we cannot publish previously copyrighted maps or satellite images created using proprietary data, such as Google software (Google Maps, Street View, and Earth). For more information, see our copyright guidelines: http://journals.plos.org/plosone/s/licenses-and-copyright.

a. You may seek permission from the original copyright holder of Figures 8 and S5 to publish the content specifically under the CC BY 4.0 license.  

Reviewers' comments:

Reviewer's Responses to Questions

**Comments to the Author**

1. Is the manuscript technically sound, and do the data support the conclusions?

Reviewer #1: Yes

Reviewer #2: Partly

Reviewer #3: Yes

2. Has the statistical analysis been performed appropriately and rigorously? 

Reviewer #1: I Don't Know

Reviewer #2: Yes

Reviewer #3: Yes

3. Have the authors made all data underlying the findings in their manuscript fully available?

Reviewer #1: Yes

Reviewer #2: Yes

Reviewer #3: Yes

4. Is the manuscript presented in an intelligible fashion and written in standard English?

Reviewer #1: Yes

Reviewer #2: No

Reviewer #3: Yes

5. Review Comments to the Author

Reviewer #1: Interesting concept study. It merits careful thinking but also raises questions regarding “regular” wearing of face covering masks in general.

Here is one main question to the authors: The Japanese society is famously known to adopt the wearing of face masks in general especially in public transportation. And this is way before COVID-19 pandemic. It is also known that KD is endemic at the highest level in Japan (followed by other eastern Asian nations). The incidence in Asian descents in their home countries as well as abroad actually supports to a great extent the genetic preposition of Asians in general and Japanese in particular to KD. The following questions therefore inherently follow: 1) how much does the face mask wearing (before COVID-19) predisposes Japanese children to attract KD from the environment perspective; 2) what is the typical school-aged (or preschool) Japanese child exposure to “polluents” and “microbes” compared to Western nations from the standpoint of wearing face masks in general (before COVID-19) – was it common but perhaps to a lesser widespread use before COVID-19 pandemic? ; 3) How do the Ig levels in Japanese children compare to Japanese descent children living in Western countries (where face masks or not common practice in normal years)?

Pleasae incorporate aspects from the above in Intro, limitations and Discussion

Back to the paper itself now.

Nice concept and interesting data source.

There should be a reviewer with deep knowledge in Public health and general populational immunology background reviewing this paper. My knowledge in KD can justifies my overall appreciation of the paper as is, but would limit me to give a sound science-based objective feed-back in the setting.

Excellent paper in my opinion in general however.

Reviewer #2: Title- Hygienic behaviors during the COVID-19 pandemic may decrease immunoglobulin G levels: implications for Kawasaki disease

In this study, Yamaguchi et al. investigated if rigorous hygiene regulations imposed during the COVID-19 pandemic affected immunity in the human population. They also investigated a potential relationship between the change in population immunity and Kawasaki disease (KD) epidemiology. Given the seriousness of the COVID pandemic, I believe such studies are essential and can add to the knowledge pool that is slowly building toward understanding the disease. The authors have done a lot of good work. However, before the manuscript is accepted for publication, authors need to address the following queries and make significant changes.

1. Foremost concern is that the manuscript is exceptionally long. The authors have included so much information in the introduction section, which ideally belongs to the discussion section. Please make the entire manuscript short and to the point. The introduction section alone has 6-7 paragraphs over-explaining everything and distracting from what they actually did and what was the rationale behind the work.

2. This study is conducted using data between January 2010 and December 2021. They mentioned that the study was approved by the ethics committee. Do they have consent from individuals whose data is used for the work? It is an important concern, and they will need to clearly define it in the M&M section.

3. The age groups authors selected for IgG analysis are infants and neonates < 0.3 years; children between 0.3 and 5 years; individuals ≥ 5 years. The rationale they gave is that IgG levels steeply surged up to 5 years of age. However, in Fig. 2a, it looks like there is almost a steady increase in IgG till > 80 years. The age groups are imbalanced. Keeping a 6-year-old child and a 70-year-old individual in the same group is not the right way of doing the analysis. Kindly break down age groups from >5 years into smaller groups.

4. In Fig. 3a, the authors show a gradual decrease in IgG from 2010 to 2021. There are years like 2016-2017, where there is a significant reduction in all age groups. What was the reason for this observation? They need to include this in the discussion as this observation is similar to COVID years. This is diluting the COVID-specific IgG decline the authors are trying to mention in their study. It is making it look like that in this region before COVID as well, there were similar observations.

5. Writing is poor, with many repetitive statements throughout the manuscript. For eg. “…the public refrain from non-urgent travel and gatherings. Schools closed.” School closed? Two work sentences? They could have merged this in the previous statement.

Reviewer #3: I read with deep interest--this manuscript entitled 'Hygienic behaviors during the COVID-19 pandemic may decrease immunoglobulin G levels: implications for Kawasaki disease" by Yamaguchi et al. The manuscript is sound; the topic is of interest; and the manuscript is carefully written.

This is among the very few manuscripts that I believe could be published in the present form.

It can be allowed to be published in its present form if its similarity check (e.g. by iThenticate or any other program) is within acceptable limits as per journal policies.

I congratulate the authors.

6. PLOS authors have the option to publish the peer review history of their article (what does this mean?). If published, this will include your full peer review and any attached files.

Reviewer #1: No

Reviewer #2: No

Reviewer #3: **Yes: **Shazi Shakil

---

## [Author Response · Author response to Decision Letter 0]

26 Aug 2022

Your reference: PONE-D-22-10882

Manuscript title: Hygienic behaviors during the COVID-19 pandemic may decrease immunoglobulin G levels: implications for Kawasaki disease

Dr Emily Chenette

Editor-in-Chief

PLOS ONE

Dr Ghulam Md Ashraf

Academic Editor

PLOS ONE

Dear Dr Chenette and Dr Ashraf,

Thank you very much for assigning the three excellent reviewers to our manuscript. We are aware that, under the present circumstances, finding experienced reviewers is increasingly difficult.

We would like to respond to the useful comments from these three Reviewers, and the Editorial Office as follows:

Comment from the Editorial Office:

Editorial comment 1: Once you have amended this/these statement(s) in the Methods section of the manuscript, please add the same text to the “Ethics Statement” field of the submission form (via “Edit Submission”)…Your ethics statement should only appear in the Methods section of your manuscript. 

Response to Editorial Comment 1: We detailed the “Ethics Statement” paragraph in the Methods section (Line 366-), as in:

Ethics statement. This study was approved by the Ethics Committee for Fukuoka Tokushukai Hospital, with reference 220302. The Ethics Committee waived the requirement for informed consent because of the retrospective nature of this study. All data were fully anonymized before the analysis.

We also amended the Ethics Statement field of the Submission form, accordingly.

Editorial Comment 2. We note that you have included the phrase “data not shown” in your manuscript.

Response to Editorial Comment 2. We deleted the phrase “data not shown”

Editorial Comment 3. We note that Figures 8 and S5 in your submission contain [map/satellite] images which may be copyrighted.

Response to Editorial Comment 3. These figures were deleted.

Editorial Comment 4. Please review your reference list to ensure that it is complete and correct. If you have cited papers that have been retracted, please include the rationale for doing so in the manuscript text, or remove these references and replace them with relevant current references. Any changes to the reference list should be mentioned in the rebuttal letter that accompanies your revised manuscript. If you need to cite a retracted article, indicate the article’s retracted status in the References list and also include a citation and full reference for the retraction notice.

Response to Editorial Comment 4. We replaced 

Y Nagao, C Urabe, H Nakamura, N Hatano. Predicting the characteristics of the aetiological agent for Kawasaki disease from other paediatric infectious diseases in Japan. Epidemiol Infect. 2016;144(3):478-492. 

with 

Y Nagao, C Urabe, H Nakamura, N Hatano. Predicting the characteristics of the aetiological agent for Kawasaki disease from other paediatric infectious diseases in Japan. Epidemiol Infect. 2016;144(3):478-492. Erratum in p. 493.

Comments from the Reviewers

Reviewer #1: Interesting concept study. It merits careful thinking but also raises questions regarding “regular” wearing of face covering masks in general.

We appreciated that Reviewer #1 found our manuscript interesting.

1) how much does the face mask wearing (before COVID-19) predisposes Japanese children to attract KD from the environment perspective; 

We have been unable to find a literature which specifically estimated the effect of face mask wearing in reducing the exposure to the KD agent(s) “before COVID-19”. However, the KD incidence decreased in 2020 substantially in Japan as compared to the pre-COVID era. This indicates that the hygienic measures (including face masks) are effective temporarily, in reducing the exposure to KD agent(s). We mentioned this in the Discussion section (Lines 241-), as in:

The KD incidence in Japan decreased by 35% in 2020 from the pre-COVID years [46], implying that the hygiene behaviors may have affected this illness. Therefore, it would be worthwhile to discuss the potential impact of the decreasing IgG on KD.

2) what is the typical school-aged (or preschool) Japanese child exposure to “polluents” and “microbes” compared to Western nations from the standpoint of wearing face masks in general (before COVID-19) – was it common but perhaps to a lesser widespread use before COVID-19 pandemic? ; 

To our knowledge, no study (at least written in English) compared microbial exposure between the Japanese and the Western people, to say nothing of children. However, we agree with the Reviewer #1 that this is an important topic which should be studied. The COVID outbreak drew attention to this topic, as we mentioned in the Introduction section (lines 49-):

In March 2020, 67% of the Japanese wore face masks in public places, at a higher rate than in Western countries (e.g. 42% in Spain, 17% in the US and 6% in the UK) [2].

3) How do the Ig levels in Japanese children compare to Japanese descent children living in Western countries (where face masks or not common practice in normal years)?

Unfortunately, we failed to identify a study which compared the IgG levels between Japanese children and Japanese-descent children in Western Countries. However, this would make an interesting research question which is relevant to the elucidation of the pathophysiology of Kawasaki Disease. We encouraged future studies toward this direction as in Lines 314- in the Discussion section: 

These heterogeneities provide an opportunity to correlate socio-behavioral changes to human immunological parameters (e.g. total IgG) and health outcomes.

Back to the paper itself now.

Nice concept and interesting data source.

My knowledge in KD can justifies my overall appreciation of the paper as is, 

Excellent paper in my opinion in general however.

We are honored by all of these positive comments from the Reviewer #1. Thank you very much.

Reviewer #2: 

Given the seriousness of the COVID pandemic, I believe such studies are essential and can add to the knowledge pool that is slowly building toward understanding the disease. The authors have done a lot of good work.

We are grateful to this Reviewer #2 for recognizing the significance of our study. Thank you very much.

1. Foremost concern is that the manuscript is exceptionally long. The authors have included so much information in the introduction section, which ideally belongs to the discussion section. Please make the entire manuscript short and to the point. The introduction section alone has 6-7 paragraphs over-explaining everything.

Following this important advice, we omitted redundant information and moved some paragraphs from the Introduction section to the Discussion section. As a result, the Introduction section was shortened into 5 paragraphs.

2. This study is conducted using data between January 2010 and December 2021. They mentioned that the study was approved by the ethics committee. Do they have consent from individuals whose data is used for the work? It is an important concern, and they will need to clearly define it in the M&M section.

We agree with this Reviewer #3 that research ethics is the most important part of a study which involves human subjects. We detailed the process of ethical clearance in Lines 366- in the Methods section as in:

Ethics statement. This study was approved by the Ethics Committee for Fukuoka Tokushukai Hospital, with reference 220302. The Ethics Committee waived the requirement for informed consent because of the retrospective nature of this study. All data were fully anonymized before the analysis.

3. The age groups authors selected for IgG analysis are infants and neonates < 0.3 years; children between 0.3 and 5 years; individuals ≥ 5 years. The rationale they gave is that IgG levels steeply surged up to 5 years of age. However, in Fig. 2a, it looks like there is almost a steady increase in IgG till > 80 years. The age groups are imbalanced. Keeping a 6-year-old child and a 70-year-old individual in the same group is not the right way of doing the analysis. Kindly break down age groups from >5 years into smaller groups.

Following this useful suggestion by Reviewer #2, we conducted an analysis which broke down the age groups (Lines 141- in the Results section):

Subdivision of the age class ≥5 years showed that, during the COVID period, the IgG declined by 42.1 mg/dL per year in 5-50 years age group (P=0.0138, n=1,325) and 42.5 mg/dL per year in ≥50 years (P=0.1521, n=952) (S2 Table).

4. In Fig. 3a, the authors show a gradual decrease in IgG from 2010 to 2021. There are years like 2016-2017, where there is a significant reduction in all age groups. What was the reason for this observation? They need to include this in the discussion as this observation is similar to COVID years. This is diluting the COVID-specific IgG decline the authors are trying to mention in their study. It is making it look like that in this region before COVID as well, there were similar observations.

We appreciated it that this Reviewer #2 read our manuscript very carefully. We mentioned this interesting observation in Lines 238- in the Discussion section:

Interestingly, there was a significant IgG decline in all age groups from 2016 to 2017 (Figure 3 a). This decrease may have been related to an increase of the face mask production in Japan, which occurred in 2015 [45]. 

5. Writing is poor, with many repetitive statements throughout the manuscript. For eg. “…the public refrain from non-urgent travel and gatherings. Schools closed.” School closed? Two work sentences? 

Following this important advice, we maximally deleted repetitive statements. The sentence “School closed” was deleted. 

Please let us thank once again to the Reviewer #2, whose comments improved our manuscript greatly.

Reviewer #3 (Dr Shazi Shakil): I read with deep interest--this manuscript... The manuscript is sound; the topic is of interest; and the manuscript is carefully written. 

This is among the very few manuscripts that I believe could be published in the present form.

It can be allowed to be published in its present form if its similarity check (e.g. by iThenticate or any other program) is within acceptable limits as per journal policies.

I congratulate the authors.

We are very grateful to all these encouraging comments from the Reviewer #3. 

Dr Shazi Shakil, Thank you very much.

---

## [Decision Letter · Decision Letter 1]

13 Sep 2022

Hygienic behaviors during the COVID-19 pandemic may decrease immunoglobulin G levels: implications for Kawasaki disease

PONE-D-22-10882R1

Dear Dr. Nagao,

We’re pleased to inform you that your manuscript has been judged scientifically suitable for publication and will be formally accepted for publication once it meets all outstanding technical requirements.

Kind regards,

Ghulam Md Ashraf, Ph.D.

Academic Editor

PLOS ONE

Additional Editor Comments (optional):

Reviewers' comments:

Reviewer's Responses to Questions

**Comments to the Author**

1. If the authors have adequately addressed your comments raised in a previous round of review and you feel that this manuscript is now acceptable for publication, you may indicate that here to bypass the “Comments to the Author” section, enter your conflict of interest statement in the “Confidential to Editor” section, and submit your "Accept" recommendation.

Reviewer #1: All comments have been addressed

Reviewer #2: All comments have been addressed

Reviewer #3: All comments have been addressed

2. Is the manuscript technically sound, and do the data support the conclusions?

Reviewer #1: Yes

Reviewer #2: Yes

Reviewer #3: Yes

3. Has the statistical analysis been performed appropriately and rigorously? 

Reviewer #1: Yes

Reviewer #2: N/A

Reviewer #3: Yes

4. Have the authors made all data underlying the findings in their manuscript fully available?

Reviewer #1: Yes

Reviewer #2: Yes

Reviewer #3: Yes

5. Is the manuscript presented in an intelligible fashion and written in standard English?

Reviewer #1: Yes

Reviewer #2: Yes

Reviewer #3: Yes

6. Review Comments to the Author

Reviewer #1: Excellent paper, looking forward to see it published soon.

ONE HUNDRED CHARACTERS REQUIRED ATTAINED

Reviewer #2: (No Response)

Reviewer #3: As far as my part of comments are concerned; I feel the manuscript is acceptable now.

I congratulate the authors.

7. PLOS authors have the option to publish the peer review history of their article (what does this mean?). If published, this will include your full peer review and any attached files.

Reviewer #1: No

Reviewer #2: No

Reviewer #3: **Yes: **SHAZI SHAKIL

---

## [Editor Report · Acceptance letter]

19 Sep 2022

PONE-D-22-10882R1 

Hygienic behaviors during the COVID-19 pandemic may decrease immunoglobulin G levels: implications for Kawasaki disease 

Dear Dr. Nagao:

I'm pleased to inform you that your manuscript has been deemed suitable for publication in PLOS ONE. Congratulations! Your manuscript is now with our production department. 

Kind regards, 

on behalf of

Dr. Ghulam Md Ashraf 

Academic Editor

PLOS ONE